# Development of a Rapid and Eco-Friendly UHPLC Analytical Method for the Detection of Histamine in Fish Products

**DOI:** 10.3390/ijerph17207453

**Published:** 2020-10-13

**Authors:** Antonello Cicero, Francesco Giuseppe Galluzzo, Gaetano Cammilleri, Andrea Pulvirenti, Giuseppe Giangrosso, Andrea Macaluso, Antonio Vella, Vincenzo Ferrantelli

**Affiliations:** 1Istituto Zooprofilattico Sperimentale della Sicilia ‘A. Mirri’, via Gino Marinuzzi, 3, 90129 Palermo, Italy; antonellocicero@gmail.com (A.C.); gaetano.cammilleri86@gmail.com (G.C.); g.giangrosso1@gmail.com (G.G.); amaca258@gmail.com (A.M.); laboratorio.residui@gmail.com (A.V.); vincenzo.ferrantelli@izssicilia.it (V.F.); 2Dipartimento di Scienze della Vita, Università degli studi di Modena e Reggio Emilia, Via Università 4, 41121 Modena, Italy; andrea.pulvirenti@unimore.it

**Keywords:** UHPLC, histamine, biogenic amine, food safety, fish products

## Abstract

We developed, validated, and confirmed with proficiency tests a fast ultra-high-performance liquid chromatography with diode array detector (UHPLC-DAD) method to determine histamine in fish and fishery products. The proposed method consists of two successive solid–liquid extractions: one with a dilute solution of perchloric acid (6%) and the second only with water. The instrumental analysis with UHPLC provides a very fast run time (only 6 min) with a retention time of approximately 4 min, a limit of quantification (LOQ) of 7.2 mg kg^−1^, a limit of detection (LOD) of 2.2 mg kg^−1^, a recovery around 100%, a relative standard deviation (RSD%) between 0.5 and 1.4, and an r^2^ of calibration curve equal to 0.9995. The method detected optimal values of the validation parameters and required a limited number of reagents in comparison to other methods reported in the literature. Furthermore, the method could detect histamine in a very short time compared with other methods. This method, in addition to being validated, precise, specific, and accurate, avoids wasting time, money, and resources, and limits the use of organic solvents.

## 1. Introduction

Fish product consumption has been steadily increasing since 1990, with a per capita consumption of 20 million tons [1]. Contextually, the global trade of fish and fish products has increased since 1986, with fish exports reaching 67.1 million tons in 2018 [1]. The incessant demand for fish products has led to an increase in control activities in order to safeguard the health of consumers. The high level of consumption of fish and fish products could lead to an increase of exposure to toxic substances such as heavy metals, persistent organic pollutants (POPs), and other contaminants [2,3,4,5,6,7]. Furthermore, several factors, such as transport and storage conditions, could lead to the formation of a food contaminant called histamine [8,9]. Histamine (2-(4-imidazolyl) ethylamine) is a biogenic amine that plays an important role in the inflammatory and allergic responses of humans [10]. Histamine is produced by proteolytic bacteria from the decarboxylation of the amino acid histidine [6,11]. The concentration of histamine in food depends on the histidine contents present and on the presence of microorganisms belonging to the genera *Vibrio, Photobacterium, Klebsiella*, and *Morganella* in tissues [12]. In predisposed individuals, even small doses (e.g., 50 mg kg^−1^) can lead to poisoning [13]. The toxicological aspects of histamine are also related to its resistance at high temperatures [14], which can compromise the safety of a product even after heat processing. The consumption of foods that contain high levels of histamine can cause scombroid poisoning [15,16]. Scombroid poisoning, also called histamine fish poisoning, is one of the most common forms of food poisoning caused by fish product consumption. Medical reporting of scombroid poisoning is frequent in Italy in relation to the high consumption of fish products [17,18]. The symptoms and treatment of this type of food poisoning are often associated with seafood allergies [15].

The pathology consists of skin redness, throbbing headache, burning mouth, abdominal cramps, nausea, diarrhea, palpitations, malaise, and rarely hyperthermia or loss of vision [15]. Symptoms usually appear within 10–30 min of the ingestion of fish and are generally self-limiting. Physical signs may include diffuse pallor, rash, tachycardia, dyspnea, and hypotension or hypertension [18,19].

A recent report of the European Authority of Food Safety (EFSA) revealed the need for a better hygiene process and other controls to prevent food poisoning caused by the consumption of food containing higher amounts of toxic biogenic amine [20]. The member states informed the EFSA that findings of certain levels of toxic biogenic amines (BA) in fermented food could be of concern, also reporting a recent increase of biogenic amine contents in some fermented foods.

The Commission Regulation (EC) No 2073/2005 on microbiological criteria for foodstuffs has established the general criteria for the analysis of fish products from fish species associated with a high amount of histidine. In particular, the recommended level of histamine in Europe was 200 mg kg^−1^ in fishery products from fish species associated with a high amount of histidine and 200–400 mg/kg in fishery products which have undergone enzyme maturation treatment in brine [21]. According to the Commission Regulation (EC) No 1441/2007, amending No. 2073/2005, the histamine content must be a maximum of 50 and 100 mg kg^−1^ in fish and fish products, respectively [22]. Regulation No. 2073/2005 specifies that the sampling plan must include nine aliquots and that high-performance liquid chromatography (HPLC) is the reference method that must be used to detect histamine presence.

Reverse-phase HPLC with a C18 column and a polar gradient has been described several times and has shown satisfactory results in several matrices [23,24,25,26]. Given this consideration, HPLC is one of the most used methods for measuring histamine levels in different biological samples [27]. There are different HPLC methods, which require different detectors to detect histamine in fish and fishery products. However, most of the methods reported in the literature use a derivatization step to improve the detection of histamine by UV or fluorescence spectroscopic detectors, and this incurs a great waste of time and/or money with an approach that is not eco-friendly [27,28,29]. Only recent findings have allowed an evaluation of the histamine content in fish products without derivatization to be achieved, but this process takes a considerable time per sample due to a sonication step of 30 min during the extraction phase [30].

Given the constant need to improve the number of routine controls to prevent histamine food poisoning caused by the consumption of fish products, the present work aimed at the development and validation of an inexpensive, fast, and eco-friendly ultra-high-performance liquid chromatography with diode array detector (UHPLC-DAD) method for the detection of histamine in fish and fish products according to European Regulations.

## 2. Materials and Methods

### 2.1. Reagents and Solution

Histamine dihydrochloride (99%), sodium 1-decanesulfonate, potassium monophosphate, potassium hydrogen phosphate trihydrate, acetonitrile, and perchloric acid (HClO_4_) were purchased from Sigma-Aldrich (Amsterdam, The Netherlands). All chemical reagents and solvents were of analytical grade. Ultrapure water was obtained from a MilliQ purification system (Merck, Darmstadt, Germany).

Phosphate buffer solution at pH 6.9 was prepared by weighing 1.7 g of potassium monophosphate, 2.85 g of potassium hydrogen phosphate trihydrate, and 0.49 g of sodium 1-decanesulfonate and dissolving these in 1 L of distilled water.

### 2.2. Standard Solutions and Calibration Curve Preparation

The histamine standard aqueous solution at 1000 mg L^−1^ was prepared by weighing 166 mg of histamine dihydrochloride and dissolving it in 100 mL of deionized water. The solution was stored at +4 °C before the analysis. The calibration standards solutions were created at concentrations of 5–20–40–80–120 mg L^−1^ by diluting standard histamine solutions at 1000 mg L^−1^ with deionized water.

### 2.3. Sample Preparation and Extraction

Forty tuna muscle samples (*Thunnus thynnus*) caught in the Mediterranean Sea (FAO Zone 37.1.3) and transported at −20 °C within 3 h to the laboratory were used as blank samples. The samples were previously homogenized and weighed (10 g) in 50 mL centrifuge test tubes, and then were spiked with a known amount of the standard histamine solution at 1000 mg L^−1^. Ten milliliters of 6% perchloric aqueous acid solution was added to the samples, and the mixture was vortexed for 1 min. Then, 30 mL of deionized water was added, and the mixture was vortexed again for 1 min. The mixture was centrifuged for 10 min at 3000 rpm, and the supernatant was transferred to a 50 mL volumetric flask and filled to the mark with deionized water. The solution was filtered on a 0.45 µm microfilter directly into the vials.

### 2.4. Ultra-High Performance Liquid Chromatography Conditions

The chromatographic separations were carried out by an Agilent 1290 UHPLC with a UV/DAD detector (Agilent Technologies, Santa Clara, CA, USA) using a Supelcosil LC-ABZ column (15 cm, 4.6 mm, DI 5 mm). The injection volume was 20 μL, the flow rate was 1.2 mL/min at room temperature, and the detector wavelength was set at 210 nm. The method involved an isocratic elution using mobile phase A, consisting of the phosphate buffer aqueous solution at pH 6.9, and mobile phase B, consisting of acetonitrile (85:15, *v*/*v*). The quantitative determination was calculated as follows:(1)$$C_{\left(C\right)}=\frac{C_{\left(s\right)}*V_{\left(F\right)}*D}{p*R}$$
where *C*_(*c*)_ is the histamine concentration on the sample (mg kg^−1^); *C*_(*s*)_ is the concentration of the analyte in the sample (mg L^−1^), which is determined by the interpolation with the calibration curve; *V*_(*F*)_ is the final volume of the sample’s extract (mL); *D* is the dilution factor; *p* is the weight of the sample (g); and *R* is the mean recovery.

### 2.5. Validation of the Analytical Method

The method was validated by an in-house validation protocol, according to Unificazione Nazionale Italiana (UNI) Comité Europè en de Normalisation (CEI EN) International Standards Organization (ISO) International Electrotechnical Commission (IEC) 17025:2018 [31]. The detection limit (LOD) and the quantification limit (LOQ) were calculated by the analysis, under repeatable conditions, of 10 independent blank samples, spiked with a constant amount of the analyte (10 mg kg^−1^). The linearity of the method was calculated by the linear regression of the areas obtained from the analysis, in triplicate, of five concentration points (5–20–40–80–120 mg L^−1^). A correlation coefficient (r^2^) value higher than 0.999 was considered acceptable for the fit of the data. The recovery and expanded uncertainty parameters were determined by fortifying blank tuna samples spiked at three concentration levels of histamine (100, 200, 400 mg kg^−1^) and performing 10 replicates for each level. The statistical analyses were carried out using R 3.2.2 software.

## 3. Results and Discussion

### 3.1. Extraction Optimization

The application of two simple and fast solid–liquid extractions, using only aqueous solutions with no solid phase extraction (SPE), as described previously [32,33,34,35], allowed the time and costs of each analysis to be reduced.

Furthermore, the method did not require a chemical derivatization process to improve the detection of the histamine by the detectors, as reported in the literature [27,28,29,36].

In fact, amines with no derivatization cannot easily be detected by HPLC because the polar structure does not allow a strong interaction with the stationary phase; ultraviolet detection can increase the analytical performance of amines analysis [37] as long as histamine has a UV-absorbing or a fluorescent group. Among derivatization reagents, o-Phthalaldehyde (OPA) is one of the most common because it is capable of reacting with amines selectively [36], and it is a fluorogenic agent that allows the analysis of histamine in HPLC-FLD. However, different factors can influence the derivatization reaction, such as the temperature and pH of the buffer [38], and often an amount of sodium sulfite is required for the reaction [38,39,40]. Other derivatizing reagents such as benzoyl and dansyl chloride create more stable derivates; however, interferences are common with these reagents because they can react with other nucleophilic groups such as alcohol [41].

By bypassing this step while maintaining high performance in terms of validation parameters, it was possible to eliminate the use of the organic solvents (acetonitrile, diethyl ether, methanol, 1,7-diaminoheptane, heptylamine, acetone, 2-mercaptoethanol, and n-heptane) or hazardous organic molecules such as o-phthalaldehyde (hazard statements H301, H314, H317, and H335) used during the extraction or the sample preparation step. The use of simple aqueous solutions instead of hazardous organic solvents allowed a very cheap and rapid method which does not compromise the operator’s health and is also eco-friendly to be achieved. These aspects are important for several reasons; for instance, there is an increase in fish product consumption [1], and this trend must be associated with an increase in food analysis to ensure product safety.

Green analytical chemistry (GAC) is a new discipline of chemistry based on 12 principles called “significance” [42] and which aims to increase the safety of operators by avoiding derivatizations and by eliminating or replacing toxic reagents. The method proposed in this work aimed to bring this analysis closer to this approach. However, the method implies the use of a reagent that is not eco-friendly, such as perchloric acid [43], which is used to precipitate protein and to extract amines from the matrix [28,44,45,46].

Further studies are needed to explore the possibility of analyzing histamine with relatively recent extraction methods, such as “Quick, Easy, Cheap, Effective, Rugged and Safe” (QuEChERS), which is used for the determination of polar and nonpolar compounds [7,47] and “dilute and shoot”, which is useful to reduce the waste of solvents during analysis [48,49], in accordance with the principles of GAC.

### 3.2. Method Optimization

#### 3.2.1. Validation Parameters

The HPLC method developed in this work showed better performance than similar methods reported in the literature [28,29,36]. Figure 1 reports the UV spectrum of histamine in the experimental conditions, showing an absorption between 200 and 240 nm and a maximum λ of 210 nm, as has already been reported numerous times in the literature [50,51,52].

The UV-DAD chromatograms at 210 nm of the standard solution, a sample spiked at 200 mg kg^−1,^ and a blank sample are shown in Figure 2.

The spiked sample is characterized by several peaks associated with the matrix and a separated peak with the retention time of 4.08 min that is attributable to histamine. The method was able to accurately measure the histamine response in the presence of any sample interference, revealing high specificity. Fish products are complex matrices, and matrix interferences have been reported in histamine analysis [13,36,53]. No interfering matrix peaks were found in the time window in which the analyte was eluted, indicating that the chromatographic conditions optimized in this method provided sufficient resolving power to distinguish the analyte from the co-eluting sample matrix compounds.

The presence of the histamine in spiked samples as verified by a double-check procedure; i.e., comparing the retention times with the standard histamine solutions and overlaying their UV-vis spectra.

It is worth noting that the retention time of histamine found with our approach (4.08 min) is the smallest compared to all literature methods that use HPLC [27,28,29,36], which have retention times between 13 and 32.5 min, which contributes to making the method very fast. Furthermore, the previous methods reported in the literature, with very long retention times, are not compatible with routine analysis, where an average of 27 samples per day were analyzed (in three batches, according to the regulations) [21,22].

The linear response of the method was verified in a range between 5 and 120 mg L^−1^ (Figure 3), with a correlation coefficient (r^2^) equal to 0.9995. This value is significantly higher than that of Tahmouzi [27] (0.9977) and in line with values reported by Piersanti and Altieri [28,29] (0.9992–0.9998).

The validation parameters obtained by the statistical analysis are shown in Table 1. The obtained values of the LOD and LOQ were 2.2 and 7.2 mg kg^−1^, respectively, which are lower than the values reported in the literature by Altieri and Piersanti [28,29], which have LOQ values equal to 10 and 50 mg kg^−1^, respectively, and are comparable to those obtained by the method of Tahmouzi [27]. A full recovery of histamine for all the validation levels was obtained; these values (100–104%) are similar to those of Altieri and Tahmouzi and higher than Piersanti (82–85%). The expanded uncertainties were within the acceptable range and are in accordance with those found by Piersanti and Altieri (10.3–15.1). Regarding the relative standard deviation (RSD) values of repeatability, it is possible to note that the method developed in this work revealed the lowest RSD (0.5–1.4) compared to the values reported in the literature (1.0–4.7) [27,28,29]. This means that, for each validation level, the standard deviation (SD), calculated based on 10 replicates, is the smallest when compared to the mean for the data set, and so the data are tightly clustered around the mean value.

The method validated in this work has proved to be more precise, in terms of recovery, than the method proposed by Nadeem et al. (2019) [30], which was based on indirect reverse-phase-HPLC without derivatization, although our work considered different concentration points in accordance with the limits imposed by the EC Regulation 2073/2005. Furthermore, our method was able to detect histamine in fish products satisfactorily without requiring an ultrasonication step, thus making it faster.

To the best of our knowledge, we can therefore assume that the method proposed in this work has been proven to be faster and more environmentally friendly than the approaches reported in the literature and also to show comparable validation results [27,29,36].

#### 3.2.2. Composition and pH of Mobile Phase

The aims of this work were to develop and validate a method according to current legislation that is green as possible and that is characterized by a fast and cheap extraction procedure. The only step which required the use of an organic solvent was the isocratic elution, which is necessary to permit a clear separation between the chromatographic peak of the analyte and the peaks relating to the matrix [54]. Nevertheless, the proportion of the organic mobile phase of the method proposed was reduced to a minimum (15%) to reduce the environmental impact and the costs of each analysis. The quantity of organic solvent used is lower than other methods reported in the literature [55,56]; this modification did not compromise the separation of histamine during the analysis. Acetonitrile is not a green solvent according to the GAC approach [57] due to irritation, air hazard, and acute toxicity [58]. However, recent studies based on the use of greener solvents than acetonitrile, such as methanol, have given unsatisfactory results [55,59].

The concentration of H^+^ in the mobile phase can affect the retention time, matrix effect, and the durability of the column, especially for pH values less than 6.0 [38,60]; an aqueous mobile phase with a pH of 6.9 allows a clear chromatogram with no matrix interferences to be obtained (Figure 2).

Amino acids are zwitterionic, and a change in pH can critically influence the quality of the chromatogram; in fact, an increase of acetonitrile in the mobile phase is not sufficient to increase the retention time of histamine and to obtain a clear separation with other amino acids in the matrix [28].

In addition to that, one of the advantages of omitting the derivation process is that some histamine-derived products are sensitive to pH, and this must be taken into account in derivation procedures because the pH has to be adjusted between the limit of histamine-derivates and the amino acids’ interfering peaks [61].

### 3.3. Application of the Method to Real Samples

The validated method was applied for the determination of histamine in fresh and processed fish samples (Table 2). One hundred and ninety-seven samples were collected from different markets of Sicily (Southern Italy) during inspections by the supervisory authorities. About 24% of the samples examined showed histamine contents, with a maximum value of 78 mg kg^−1^. Among the processed fish samples, only the anchovy paste showed the presence of histamine.

### 3.4. Proficiency Test

Proficiency tests (PTs) represent a tool to test the performance of laboratories and to maintain the quality of analysis in accordance with ISO/IEC 17025:2018. The method proposed was tested by a proficiency test provided by FAPAS^®^ (FCAL10-SEA7 in canned fish as matrix), giving a z-score of −0.2. Other two proficiency tests from Test Veritas S.r.l, (T1930A1 and T1930A2, consisting of tuna muscle and canned tuna, respectively) were carried out, giving z-scores of 0.39 and −1.83, respectively. The z-scores obtained guaranteed satisfactory reproducibility, given the fact that the results obtained were within the limit of ±2.00. These results are comparable with other methods that were reported previously and which require a derivatization step [36,59].

## 4. Conclusions

A very fast, isocratic UHPLC method using a UV-DAD detector was developed to determine histamine in fish products. The method uses two fast solid–liquid extractions with aqueous solutions and a simple composition of the mobile phase with a minimum percentage of the organic component (15%).

HPLC with acetonitrile/methanol is commonly used for histamine analysis, and with the developed method, it is possible to analyze histamine without increasing the costs of analyses or changing the solvents.

These features combined with the fast UHPLC run time (6 min) allow us to use this method for the routine analysis of fresh and processed fish products, as shown by the proficiency test analysis.

Considering the money–time costs of the derivatization procedure, the pH adjustment required, and the impact on pollution that derivatizations have, the development of a method without this analytical step is the first step on the path of green analytical chemistry.

We can therefore conclude that the proposed method achieved the aim of realizing and validating an easy, inexpensive, fast, and almost eco-friendly UHPLC analytical method that is precise, sensitive and specific.

## Figures and Tables

**Figure 1 ijerph-17-07453-f001:**
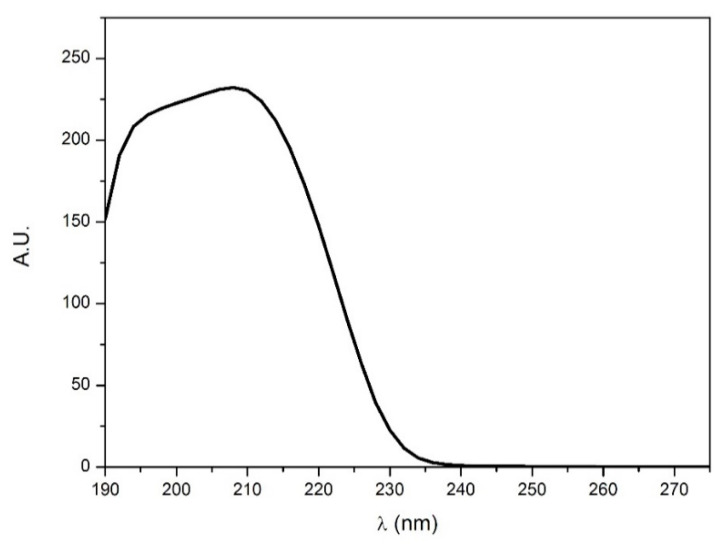
UV-Vis spectrum of a histamine standard aqueous solution at 40 mg L^−1^.

**Figure 2 ijerph-17-07453-f002:**
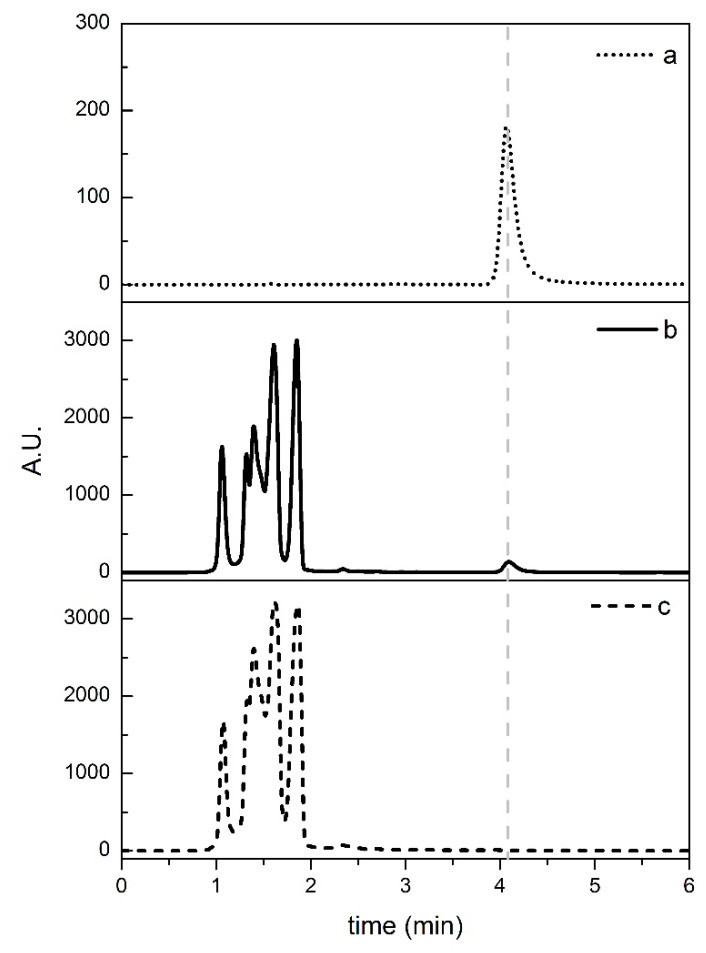
UV-diode array detector (DAD) chromatograms at 210 nm. (**a**) Standard solution of histamine at 40 mg L^−1^; (**b**) chromatogram of a sample spiked with histamine 200 mg kg ^−1^); (**c**) chromatogram of a blank sample. The grey line at 4.08 min is a visual aid.

**Figure 3 ijerph-17-07453-f003:**
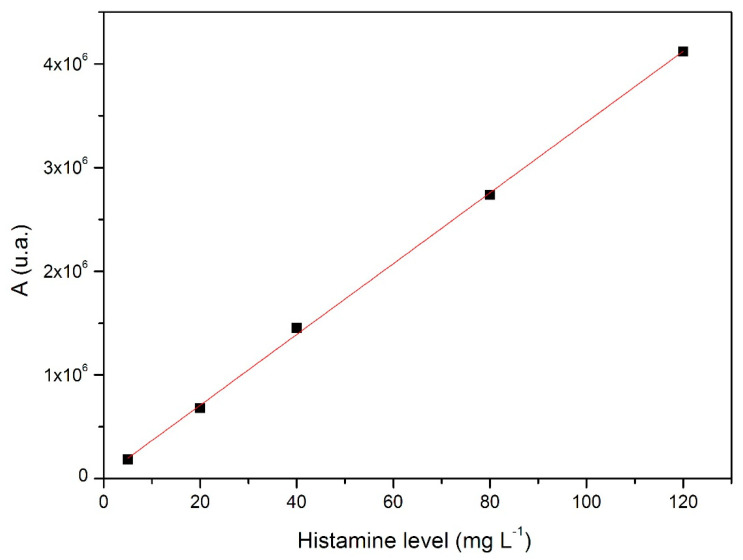
Linearity results of the method proposed for the considered histamine concentration range.

**Table 1 ijerph-17-07453-t001:** Results obtained by the validation process. RSD: relative standard deviation.

Histamine Level (mg kg^−1^)	Mean ± SD (mg kg^−1^)	RSD (%)	Expanded Uncertainty (%)	Recovery (%)
100	104.0 ± 0.9	0.9	16	104
200	200 ± 3	1.4	9.5	100
400	401 ± 2	0.5	5	100

**Table 2 ijerph-17-07453-t002:** Fresh and processed fish samples analyzed by the validation process. LOD: limit of detection.

Sample Type	N	>LOD (%)	Mean ± SD (mg kg^−1^)
Bluefin tuna fillets	42	6 (14.3)	14.62 ± 79.44
Anchovies	27	2 (7.4)	43.12 ± 10.5
Mackerel	27	6 (22.2)	5.26 ± 9.8
Mackerel in oil	37	-	-
Tuna in oil	37	-	-
Anchovy paste	27	9 (33.3)	20.86 ± 29.50

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
