# Peer review of "Development of a Rapid and Eco-Friendly UHPLC Analytical Method for the Detection of Histamine in Fish Products"

_ijerph, 2020, doi:10.3390/ijerph17207453_

Round 1
Reviewer 1 Report
line 29-31 - Sentence is ambiguous. Also what is the last one at the end of the sentence?
line 38-40 - Recast the sentence and make it clearer.
line 44 - What is prevent histamine? The whole sentence is incomplete.
Look at my comments on the manuscript. Lines 65, 74, 133-134.

Reviewer 2 Report
This manuscript by Cicero et al describes a novel procedure for the detection of histamine in fish products by extraction from muscle tissue followed by UHPLC analysis. The authors describe the sensitivity of the assay as well as its correspondence with histamine concentration in samples spiked with exogenous histamine and show clearly that the spiked histamine can be detected without the need for derivatization. There are some issues with the presentation, particularly in terms of English grammar, which should be improved, but in terms of worthiness of publication there are two important points which must be addressed:
1) How does this assay compare to the kits which are currently commercially available? The authors discuss their results in comparison to similar methods involving chemical derivatization of histamine but no meaningful comparison is made with commercially available products such as those from Romer labs or Cell Biolabs. The authors tout the speed and low cost of this procedure but a column based assay which requires 96 samples to be run in serial may have significant disadvantages in terms of time and labor cost when compared with a plate-based assay which runs 96 in parallel. These comparisons are essential for understanding the applicability of this assay for current practice.
2) How well does this extraction procedure work for samples with "naturally occuring" histamine? The ability to extract exogenously-added histamine from muscle tissue does not, in and of itself, imply the ability to extract histamine in the native context. I am unfamiliar with how one might expect histamine to be localized in fish, but even we accept that the concentration in muscle is relevant for, say, an entire canned sardine, intracellular histamine might be much more difficult to extract than histamine added directly to the homogenized tissue.
Reviewer 3 Report
This manuscript presents a valuable and carefully documented piece of work on the HPLC analytics of histamine in fish and fish products.
The article submitted requires extensive improvement as exemplified below.
The English is incomprehensible at several instances. To point out some examples, and provide suggestions for more apt phrasing, on lines 12-13 it is not clear what was validated and confirmed (the 'present work' or the UHPLC-DAD method ?). "UHPLC", "DAD", "LOQ", "LOD" should be defined at first occurrence (lines 13, 17). On line 14, 'consist' is in plural but 'method' is in singular, and similarly on lines 15-16, 'analysis' is in singular but 'provide' is in plural. Overall, the singular and plural forms should be changed to fit each other. The general reader cannot know what LOQ, LOD, or RSD are. On line 17, a capital 'K' appears twice; the latter, for a chemist, usually refers to potassium – the abbreviation for 'kilo' is 'k'. The above abbreviations (UHPLC, DAD, etc.) should be presented in unabridged form where they first appear in the article.
On line 26, what does 'direct' human consumption mean ? On line 41, change "poisoning condition" to "toxic conditions". On line 42, what is meant by the 'richness' in histidine? or 'the assumption' of foods ? On line 46,"assumption" must be changed to "consumption". Furthermore, please clarify how the most common form of intoxication in Italy is due to histamine fish poisoning (lines 47-48 and 56; in any case, "intoxication" here presumably means "food poisoning"). On line 95, it is not clear what "fortified" means here. On the next line, note that the abbreviation for milliliter is mL. On line 128, change 'necessaries' to 'necessary'. Amines, derivatized or not, of course do not 'detect' in HPLC (line 130) but rather 'are detected'. Line 134 should read 'capable of reacting with amines selectively [39] and it is a fluorogenic reagent that allows the analysis ..." On line 135, change 'parameters' to 'factors, and line 146, change 'resulting' to 'and are'. On line 151, what is the purpose of the words '...twelve principles (SIGNIFICANCE')' other than the fact that there are 12 letters in it? On line 160, change 'respect' to 'observes'. On line 176, change 'one separated peak in' to 'a separate peak with'. On lines 177-179, the authors mention sample interferences but do not explain what they are and how their new method obviates such interferences. On line 203 the authors mention that their work has revealed the lowest RDS values compared to those in the lit. but they of course mean RSD values.
In the list of references, almost all Latin names of fish, microorganisms etc. appear in the non-italicized form.
When all the unclear, meaningless or erroneous words and phrases in this MS are replaced by the appropriate expressions, the result may very well be a piece of research well worth publication.
Round 2
Reviewer 2 Report
I think the extension of this method to measure histamine under real-world conditions establishes the relevance of the results presented.
Reviewer 3 Report
Please finds enclosed a pree-review-8919850.v1.pdf file.
